# Multiscale Neural PDE Surrogates for Prediction and Downscaling: Application to Ocean Currents

## Abstract

Accurate modeling of physical systems governed by partial differential equations is a central challenge in scientific computing. In oceanography, high-resolution current data are critical for coastal management, environmental monitoring, and maritime safety. However, available satellite products, such as Copernicus data for sea water velocity at $\sim 0.08°$ spatial resolution and global ocean models, often lack the spatial granularity required for detailed local analyses. In this work, we (a) introduce a supervised deep learning framework based on neural operators for solving PDEs and providing arbitrary resolution solutions, and (b) propose downscaling models with an application to Copernicus ocean current data. Additionally, our method can model surrogate PDEs and predict solutions at arbitrary resolution, regardless of the input resolution. We evaluated our model on real-world Copernicus ocean current data and synthetic Navier–Stokes simulation datasets.

## 1 Introduction

Accurate and high-resolution marine current fields are foundational to numerous marine applications, coastal engineering design, and autonomous navigation. Datasets like Copernicus ocean analysis product Copernicus Marine Service (2025) provide global coverage at roughly $0.08° \times 0.08°$ (approximately 9 km in mid-latitudes), which are insufficient for applications requiring detailed local dynamics.

Downscaling methods, both dynamical and statistical, have been used to bridge this resolution gap. While dynamical downscaling, employing regional ocean models, is physically rigorous, it demands substantial computational resources, often requiring days of runtime on HPC clusters. Statistical approaches offer computational efficiency but struggle with the multiscale and non-linear nature of fluid dynamics Kruyt et al. (2022).

Deep learning has emerged as an alternative to traditional statistical methods, since it can learn from data the mappings from coarse- to fine-scale representations. Initial efforts with CNNs and GANs achieved success in meteorology (Vosper et al., 2022) and image-based super-resolution (Dong, 2015). However, these models typically require fixed upsampling factors and lack fidelity when generalizing to unseen resolutions or evolving physical dynamics.

Neural operators, particularly Fourier Neural Operators (FNOs) (Li et al., 2020), Transolver (Wu et al., 2024), FactFormer (Li et al., 2023), and Latent Spectral Models (LSMs) (Wu et al., 2023), have demonstrated remarkable ability to learn resolution-agnostic operators governing PDEs. FNOs have been successfully applied to atmospheric and oceanographic forecasting (Sun et al., 2024).

Yang et al. (2023) applied FNOs for climate model downscaling and to solve partial differential equations (PDE) that are approximated with another numerical solver. In this work, we generalize this model to timeseries, solving PDEs at arbitrary resolutions at the same time. Further, we evaluate our method in a downscaling task with practical importance such as the Copernicus ocean current dataset.

Yang et al. (2023)'s DFNO model addressed downscaling for climate data and PDE solutions at arbitrary resolutions, where the PDE solution is generated via numerical solvers, not the model itself. Our work extends this paradigm in two significant directions. First, we generalize their model to

handle temporal sequences, enabling the prediction of PDE solutions at arbitrary spatial resolutions using the same model. Second, we benchmark multiple downscaling models inspired by the DFNO by applying it to the Copernicus ocean current dataset for static downscaling, to demonstrate its real-world impact geophysical data.

Our main contributions are as follows.

- We benchmark multiple models for arbitrary-resolution downscaling and apply them to physical observations that need downscaling (ocean current from Copernicus marine data).
- We develop a surrogate model capable of predicting PDE solutions at arbitrary resolutions-independent of the input resolution, giving more flexibility and extent to the model.

To the best of our knowledge, this is the first work to present a neural surrogate capable of accurate, resolution-agnostic prediction, where the output resolution does not depend on the input resolution.

## 2 RELATED WORK

### 2.1 NEURAL OPERATORS

Neural operators are models that learn mappings between infinite-dimensional function spaces and have recently emerged as powerful tools to approximate solution operators of partial differential equations (PDEs) (Li et al., 2020; Wu et al., 2024; Li et al., 2023). These models achieve strong performance on a variety of parametric PDE benchmarks, including Navier-Stokes, the Darcy flow, and Burgers' equation, while offering orders-of-magnitude speed-ups over classical numerical solvers. The Fourier Neural Operator (FNO) (Li et al., 2020) performs operator learning in Fourier space, enabling efficient global convolution. Latent Spectral Models (LSMs) project high-dimensional PDE fields into lower-dimensional latent spaces, where the equations are solved to improve both accuracy and computational efficiency for fluid and solid mechanics (Wu et al., 2023). Transolver (Wu et al., 2024) introduces physics-informed attention mechanisms, enabling the learning of PDE dynamics on unstructured meshes and complex geometries, thus reducing discretization dependence and surpassing previous neural operator architectures.

Neural operators are increasingly applied in oceanography. For example, Chattopadhyay (2023) proposed OceanNet, a hybrid FNO and predictor–evaluate–corrector model that learns Gulf Stream circulation dynamics and achieves up to $5 \times 10^5$ times speedup over classical numerical ocean models.

### 2.2 EMBEDDING PHYSICS IN DEEP LEARNING

In physics-based applications, it is critical that neural network outputs not only approximate ground truth but also remain consistent with the governing physical laws, which is essential for downstream applications and model trustworthiness. Incorporating physical priors into neural models has been shown to better capture observed physical properties. Techniques such as soft and hard constraint losses have been applied in atmospheric emulation, where physics-constrained models achieve lower errors while maintaining fidelity to the underlying equations (Beucler et al., 2021; Daw, 2020). Moreover, Yang et al. (2023) demonstrated that the introduction of physics-informed constraint layers further enhances fidelity and reduces error in climate downscaling tasks (Yang et al., 2023; Harder et al., 2024).

### 2.3 ARBITRARY-RESOLUTION DOWNSCALING

Conventional neural network downscaling models, which operate between finite-dimensional spaces, are typically limited to fixed input and output sizes. As a result, a single trained model can only downscale inputs with a predefined upsampling factor; that is, the output resolution must match the resolution anticipated during training. For example, CNN-based methods have been used to downscale meteorological fields such as wind (Campbell et al., 2025), precipitation (Vosper et al., 2022), and solar radiation, often employing multistep cascades to achieve high-resolution output. However, these models exhibit degraded performance when applied to unseen upsampling factors.

To address this limitation, Yang et al. (2023) introduced an FNO-based zero-shot downscaling model that generalizes to arbitrary resolutions without retraining. This approach outperforms both traditional

super-resolution models and conventional neural PDE solvers on Navier–Stokes simulations and ERA5 climate fields. However, it is important to note that, in the PDE setting, the model only performs downscaling on solutions generated by external numerical solvers. *Our goal is to overcome this limitation by developing a neural operator that (a) directly solves PDEs and (b) generates solutions at arbitrary resolution, independent of input resolution.*

# 3 METHODOLOGY

## 3.1 DATA SOURCES

To evaluate our proposed model, we have considered as data sources Navier-Stokes data and satellite observations of the ocean currents.

**Navier-Stokes** We used synthetic velocity fields based on the 2D incompressible Navier–Stokes equations in vorticity form on the periodic unit torus $\Omega = (0, 1)^2$:

$$\partial_t \omega(x, t) + \mathbf{u}(x, t) \cdot \nabla \omega(x, t) = \nu \Delta \omega(x, t) + f(x), \tag{1}$$

$$\nabla \cdot \mathbf{u}(x, t) = 0, \tag{2}$$

$$\omega(x, 0) = \omega_0(x), \tag{3}$$

where $\omega$ is the scalar vorticity, $\mathbf{u}$ is the velocity field, and $\nu = 10^{-4}$ is the viscosity. The velocity is recovered from the vorticity via the stream function $\psi$, using:

$$\mathbf{u} = (\partial_y \psi, -\partial_x \psi), \quad -\Delta \psi = \omega. \tag{4}$$

Following Li et al. (2020), the forcing term is fixed as $f(x) = 0.1 \left( \sin(2\pi(x_1 + x_2)) + \cos(2\pi(x_1 + x_2)) \right)$, and the initial vorticity $\omega_0(x)$ is sampled from a Gaussian random field with mean zero and spectral decay: $\omega_0 \sim \mathcal{N}(0, r^{3/2}(-\Delta + 49I)^{-2.5})$.

A total of 10,000 simulations were run at a spatial resolution of $64 \times 64$, using randomly initialized conditions. Each simulation was evolved over 50 time steps with a fixed viscosity of $10^{-4}$. The dataset was split into 7,000 training samples, 2,000 validation samples, and 1,000 test samples. For each time step, we also constructed lower-resolution versions of the data by applying average pooling to obtain $32 \times 32$ and $16 \times 16$ grids, we gave 5 time steps as input and predicted the next 5 timesteps. The final dataset contains both the full-resolution solutions and their down sampled counterparts from the same timesteps and a window of 5 consequent timesteps.

**Copernicus Data** Real-world ocean current data was obtained from the Copernicus Marine Environment Monitoring Service (CMEMS). The dataset consists of global ocean surface velocities at 0.08° spatial resolution (~8 km), providing northward and eastward velocity components. This data combines satellite altimetry, in situ observations, and numerical ocean models through data assimilation. We selected regional subsets covering different oceanographic regimes to evaluate the generalization of the model in varying flow characteristics and coastal dynamics. We split the data into $128 \times 128$ patches. The dataset was then divided into 800 training samples, 200 validation samples, and 100 test samples.

## 3.2 RESOLUTION-AGNOSTIC NEURAL OPERATOR FRAMEWORK FOR DOWNSCALING

We propose a flexible framework for simultaneous PDE solution prediction and downscaling, applicable to both temporal and static settings. The general architecture, illustrated in Figure 1, begins with a low-resolution input that optionally undergoes preprocessing—such as gradient transformation in gradient-based models—followed by a neural network and an upsampling block. The resulting output is then processed by a neural operator. Finally, a reconstruction block and an optional physical constraint layer complete the pipeline.

For the specific task of ocean current downscaling, we evaluated several model variants within this framework. We first introduced **DUNO**, which uses a U-shaped Neural Operator (UNO) in the neural

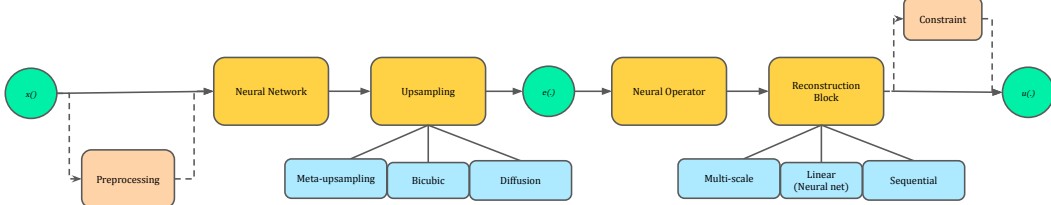

Figure 1: The figure (inspired by Yang et al. (2023)) shows the overall structure of our Temporal/static downscaling model. The low-resolution input a goes through an optional preprocessing (gradient transformation for gradient based methods) then a neural network and an upsampling block. Then an embedding function $e()$ is returned. Finally, a neural operator takes in $e()$ and outputs a function which gets into a reconstruction block and an optional constraint layer.

operator block. UNO generalizes well across different PDE types and is more expressive than the standard FNO. the rest of the framework is similar to the DFNO Yang et al. (2023).

We also introduce the **SpecDFNO** which extends the standard DFNO architecture by introducing a second neural operator that predicts the residual between the initial FNO output and the ground truth. Inspired by Qin et al. (2024), this residual is then added to the base prediction, enhancing the model's ability to capture high-frequency components often lost in downscaling.

Extending this further, the **SpecDFNO with Diffusion-Based Upsampling (SpecDFNODiff)** replaces the explicit upsampling operation with a learned generative diffusion prior. The diffusion process is conditioned on the low-resolution input, allowing the model to generate high-resolution fields that are spatially coherent and physically plausible.

We also explored gradient-based strategies. The **MetaGradDFNO** model applies the DFNO architecture on gradient fields derived using Sobel filters (applied in the preprocessing block). This model also uses a meta-learning mechanism (in the upsampling block) that learns a weighted combination of nearest-neighbor, bilinear, and bicubic interpolation kernels, enabling context-aware upsampling.

Complementary to this, the **Multiscale Gradient DFNO (MultiGradDFNO)** captures structural information across multiple scales by using parallel convolutional branches with varying kernel sizes to process gradient fields (in the reconstruction block). These branches are then merged via a convolution, which helps retain rich spatial features present even at coarse resolution.

Across all models, physical conservation laws are softly enforced through a softmax constraint layer, a mechanism demonstrated to be effective in geophysical settings by Harder et al. (2024).

### 3.3    Models for Direct PDE Solution Prediction at Multiple Scales

In addition to downscaling tasks, we adapt our architecture for direct prediction of PDE solutions directly at multiple resolutions. The models **Temp_DFNO** and **Temp_SpecDFNO** extend the DFNO and SpecDFNO architectures by incorporating a temporal dimension. In these versions, the convolutions are performed not only across spatial axes but also along the temporal axis, allowing the network to capture spatiotemporal dynamics inherent in time-evolving PDE systems.

### 3.4    Training Strategy

**Temporal Modeling (for PDE Surrogate Task):**    The model receives a sequence of five consecutive low resolution frames ($16 \times 16$) as input and predicts the next five frames at both low ($16 \times 16$) and high ($32 \times 32$) resolutions. The zero shot evaluation is performed on the $64 \times 64$ output from the $16 \times 16$ input.

**Benchmarking Static Downscaling on Copernicus Data:**    Models are evaluated on the task of static downscaling using Copernicus current marine data. Inputs consist of low-resolution velocity fields, and models predict high-resolution outputs ($2\times$, $4\times$ and $8\times$ downscaling). Regional subsets representing different oceanographic regimes are used to assess generalization.

As baselines, we used CNNx2 and CNNx4 models. Each is implemented as a four-level U-Net, with increasing feature dimensions at each level (64, 128, 256, 512). Each encoder level employs a double convolution block comprising two 3×3 convolutional layers with batch normalization and ReLU activation, followed by 2×2 max pooling for spatial downsampling. The decoder mirrors the encoder structure with transposed convolutions for upsampling and skip connections to preserve fine-grained spatial information. CNNx2 and CNNx4 refer to training with 2× and 4× downsampling, respectively. For evaluation on both 2 times and 4 times downscaling. The 2 times downscaling outputs by CNN-2 increase their resolution to 4 times through model recursion and bicubic interpolation. The 4 times downscaling outputs by CNN-4 decrease their resolution to 2 times through average pooling and bicubic interpolation.

**Loss Functions and Normalization:** We employ both L1 (MAE) and L2 (MSE) losses during training and evaluation. For perceptual quality assessment, PSNR and SSIM metrics are also computed (details provided in the Appendix). Additionally, input channels are normalized independently using channel-wise normalization.

## 4 RESULTS

### 4.1 TEMPORAL MODELS FOR MULTISCALE PDE SOLVING

Figure 2: Low-resolution input (row 1), predictions at $16 \times 16$ and $32 \times 32$ (rows 2 and 4), and corresponding ground truth (rows 3 and 5) for 2D Navier–Stokes.

We first evaluate the performance of the model on the 2D incompressible Navier-Stokes dataset. The goal is to learn a surrogate that accurately simulates spatio-temporal dynamics across multiple spatial resolutions. Results for two DFNO variants—standard (**Temp_DFNO**) and residual-based (**Temp_SpecDFNO**)—are reported in Table 1.

These results underscore the effectiveness of our proposed architectures as resolution-agnostic PDE surrogates, capable of simulating fluid flow dynamics with downscaling performance comparable to DFNO2 and DFNO4 from Yang et al. (2023), while removing the dependency on a separate numerical solver.

Table 1: Performance of Temporal_DFNO, Temporal_SpecDFNO, and DFNO models at different resolutions. DFNO-2 and DFNO-4 values are taken from Yang et al. (2023); PSNR and SSIM for these models are not provided.

| Model | Resolution | MAE | MSE | PSNR | SSIM |
|---|---|---|---|---|---|
| Temp_DFNO | 16x16 | 0.017941 | 0.000603 | 33.82 | 0.9920 |
| | 32x32 | 0.017426 | 0.000573 | 34.05 | 0.9878 |
| | 64x64 | 0.019775 | 0.000722 | 32.64 | 0.9805 |
| Temp_SpecDFNO | 16x16 | 0.017806 | 0.000599 | 34.00 | 0.9915 |
| | 32x32 | 0.017372 | 0.000568 | 34.15 | 0.9882 |
| | 64x64 | 0.019736 | 0.000712 | 32.76 | 0.9811 |
| DFNO-2 | 32x32 | 0.0124 | 0.0004 | – | – |
| | 64x64 | 0.0246 | 0.0018 | – | – |
| DFNO-4 | 32x32 | 0.0208 | 0.0012 | – | – |
| | 64x64 | 0.0168 | 0.0007 | – | – |

### 4.2 DOWNSCALING COPERNICUS OCEAN CURRENT DATA

Despite the absence of temporal information, the DFNO maintains coherent internal structures and sharp gradients better than other baselines.

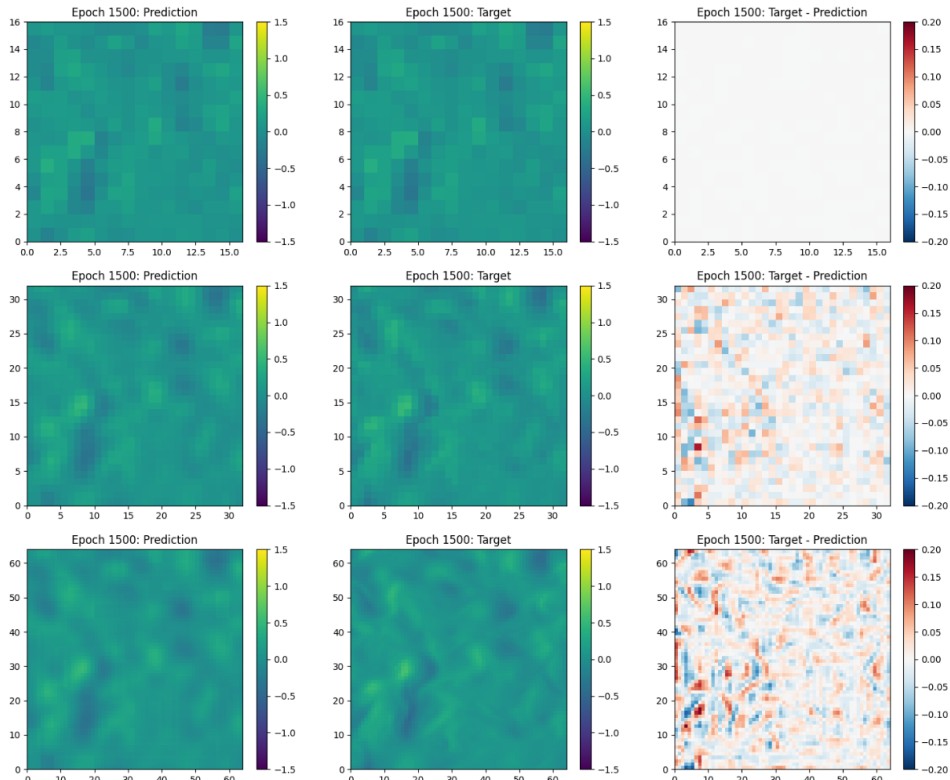

Figure 3: Ground truth vs. predictions of DFNO. Rows correspond to different output resolutions: $16 \times 16$, $32 \times 32$ and $64 \times 64$. The first column shows the model predictions, the second shows the ground truth, and the third displays the difference between them.

To evaluate the performance of our models on real-world oceanographic data, we used sea surface velocity fields provided by the Copernicus Marine Environment Monitoring Service. To simulate coarse observations, we downsampled the original fields using average pooling to generate inputs at resolutions corresponding to 2×, 4×, and 8× coarsening factors. The model was then tasked with reconstructing higher-resolution fields from the lowest-resolution inputs. Unlike the synthetic Navier–Stokes dataset, no temporal supervision was used; each sample was treated independently as a static snapshot.

## 5  DISCUSSION

### 5.1  MODEL COMPARISON

#### 5.1.1  PDE SOLVING AT MULTIPLE RESOLUTIONS

Our temporal DFNO models demonstrate strong predictive capability across multiple resolutions on the Navier–Stokes benchmark. These models not only match, but in some cases outperform the DFNO2 and DFNO4 models of Yang et al. (2023), which rely on external numerical solvers to generate low-resolution solutions before downscaling them. In contrast, our approach integrates solution generation and resolution-agnostic inference into a single neural operator framework. Thus, the proposed models has successfully combined the predictive power of a classic numerical solver as well as the physics-based downscaling ability of the DFNO.

Table 2: Downscaling results at $32 \times 32$ and $64 \times 64$ .

| Model | Loss | 32×32 | | | | 64×64 | | | |
|---|---|---|---|---|---|---|---|---|---|
| | | MAE | MSE | PSNR | SSIM | MAE | MSE | PSNR | SSIM |
| CNN_2x | L1 | 0.424 | 0.19343 | 14.06813 | 0.10542 | 0.4246 | 0.1945 | 15.3113 | 0.0992 |
| | L2 | 0.424 | 0.18774 | 14.19797 | 0.12992 | 0.4244 | 0.1890 | 15.4370 | 0.1123 |
| CNN_4x | L1 | 0.3954 | 0.16950 | 14.64186 | 0.12584 | 0.3953 | 0.1720 | 15.8440 | 0.1210 |
| | L2 | 0.395 | 0.16639 | 14.72229 | 0.12872 | 0.3954 | 0.1663 | 15.9890 | 0.1163 |
| DFNO | L1 | 0.02134 | 0.00099 | 36.97443 | 0.96785 | 0.04057 | 0.01749 | 33.15377 | 0.85038 |
| | L2 | 0.01822 | 0.00059 | 39.23959 | 0.97699 | 0.04151 | 0.01640 | 33.23573 | 0.84019 |
| DUNO | L1 | 0.03501 | 0.00254 | 32.88501 | 0.91708 | 0.04271 | 0.00320 | 32.18915 | 0.82181 |
| | L2 | 0.03440 | 0.00217 | 33.57706 | 0.92083 | 0.04251 | **0.00314** | 32.55057 | 0.82200 |
| MetaGradDFNO | L1 | 0.01424 | 0.00045 | 40.37440 | 0.98574 | 0.05493 | 0.00962 | 30.33814 | 0.73601 |
| | L2 | **0.01370** | **0.00033** | **41.74495** | 0.98638 | 0.05685 | 0.00851 | 30.10898 | 0.71987 |
| MultiGradDFNO | L1 | 0.01465 | 0.00049 | 40.07315 | 0.98396 | 0.05513 | 0.00851 | 30.04188 | 0.73671 |
| | L2 | 0.01380 | 0.00034 | 41.58843 | **0.98690** | 0.05672 | 0.00802 | 29.83308 | 0.71801 |
| SpecDFNO | L1 | 0.03407 | 0.00242 | 33.09110 | 0.92286 | 0.04283 | 0.00682 | 32.23081 | 0.82589 |
| | L2 | 0.02451 | 0.00106 | 36.66621 | 0.95871 | **0.03653** | 0.00655 | **34.16133** | **0.87433** |
| SpecDFNODiff | L1 | 0.01836 | 0.00064 | 38.85981 | 0.97519 | 0.03796 | 0.00367 | 33.97083 | 0.86536 |
| | L2 | 0.01712 | 0.00049 | 40.07214 | 0.97818 | 0.03879 | 0.00355 | 33.90497 | 0.86177 |

Table 3: Downscaling to $128 \times 128$

| Model | Loss | MAE | MSE | PSNR | SSIM |
|---|---|---|---|---|---|
| CNN_2x | L1 | 0.424 | 0.1952 | 15.8162 | 0.09986 |
| | L2 | 0.424 | 0.1895 | 15.944 | 0.105 |
| CNN_4x | L1 | 0.395 | 0.1723 | 16.358 | 0.1176 |
| | L2 | 0.395 | 0.1666 | 16.504 | 0.111 |
| DFNO | L1 | 0.06488 | 0.00936 | 29.00970 | 0.60895 |
| | L2 | 0.06488 | 0.00936 | 29.00905 | 0.60892 |
| DUNO | L1 | 0.06199 | 0.00797 | 29.70524 | 0.62467 |
| | L2 | **0.05851** | 0.00755 | 29.94049 | **0.65233** |
| MetaGradDFNO | L1 | 0.06477 | 0.00934 | 29.01829 | 0.60980 |
| | L2 | 0.06479 | 0.00935 | 29.01331 | 0.60976 |
| MultiGradDFNO | L1 | 0.06488 | 0.00894 | 29.20734 | 0.60906 |
| | L2 | 0.06486 | 0.00936 | 29.00910 | 0.60886 |
| SpecDFNO | L1 | 0.06443 | 0.00923 | 29.06780 | 0.61219 |
| | L2 | 0.05971 | **0.00748** | **29.98298** | 0.64175 |
| SpecDFNODiff | L1 | 0.06052 | 0.00752 | 29.96232 | 0.63049 |
| | L2 | 0.06386 | 0.00788 | 29.75918 | 0.60965 |

### 5.1.2 STATIC DOWNSCALING OF REAL-WORLD OCEAN DATA

On Copernicus sea surface velocity fields, DFNO variants significantly outperform conventional CNN baselines at all downscaling levels. However, as the resolution gap increases (at 8×), performance deteriorates (Table 3) due to the lack of informative coarse-scale details.

**2× downscaling** (16×16 → 32×32): All neural operator variants substantially outperform CNN baselines. MetaGradDFNO achieves the best overall performance. The dramatic performance gap between neural operators and CNNs (MAE improvement of 30×) demonstrates the importance of spectral representations for fluid flow reconstruction.

**4× downscaling** (16×16 → 64×64): Performance degradation becomes evident as the reconstruction task becomes more challenging. SpecDFNO emerges as the most robust and consistent model, while gradient-based variants show reduced effectiveness at this scale and later scales. Interestingly, DUNO maintains competitive MSE performance.

**8× downscaling** (16×16 → 128×128): Significant performance degradation occurs in all models. DUNO demonstrates good performance, and SpecDFNO and SpecDFNODiff are overall and perceptually better.

**Analysis**

**Gradient-Enhanced Models**  : MetaGradDFNO and MultiGradDFNO excel at moderate downscaling factors, leveraging multiscale information. However, their performance diminishes at higher ratios, where structural details become increasingly sparse.

**Spectral Residual Methods**  : SpecDFNO shows robust performance across all scales, particularly excelling at zero-shot 4× downscaling. The diffusion-enhanced variant (SpecDFNODiff) provides marginal improvements but with an increased computational overhead.

### 5.1.3 Limitations and Scaling Behavior

While our models show robust performance at moderate resolution increases (2× and 4×), their accuracy degrades with higher downscaling factors (e.g., to 128×128). In these cases, models tend to oversmooth outputs or hallucinate details.

This degradation reflects physical reality: As resolution increases, unresolved subgrid physics (e.g., turbulence, stratification, and nonlinear instabilities) becomes dominant. The coarse input data no longer contain sufficient information to accurately infer high-resolution dynamics.

### 5.2 Future Work

The previously discussed downscaling limitations of our models suggest two immediate directions for future research. One direction is to further improve the downscaling capability of both the PDE surrogate models and the static downscaling models applied to Copernicus sea velocity data. Another important avenue is to theoretically characterize the limits of these models, particularly as the governing physical behavior and equations change with increasing resolution. Incorporating uncertainty quantification through probabilistic neural operators and ensemble-based diffusion strategies could also help express confidence in high-resolution outputs, especially in underdetermined or data-sparse regimes.

## 6 Conclusion

We show that neural-operator downscaling can deliver higher-resolution current maps from low-resolution inputs, with strong gains over CNN baselines for Copernicus currents and *accurate multi-resolution temporal predictions without external solvers*. These results position neural operators as practical, scalable tools for ocean current analysis, and motivate uncertainty-aware and theory-driven extensions to safely push beyond moderate downscaling.

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

## A  TRAINING CONFIGURATION

### A.1  OPTIMIZER AND HYPERPARAMETERS.

We train all models using the Adam optimizer with an initial learning rate of $1 \times 10^{-3}$. Each model is trained for 600 epochs with a batch size of 16.

### A.2  DATA PREPROCESSING

- **Normalization:** Z-score normalization is applied per velocity component (northward and eastward) independently.
- **Train/Validation/Test Split:** The datasets are split using a 70%/15%/15% ratio.

## B  VISUALIZATIONS

Here we provide some visualization for the downscaling PDE solver and the benchmarking of DFNO variants on Copernicus data.

### B.1  DOWNSCALING AND PREDICTING PDE SOLUTIONS

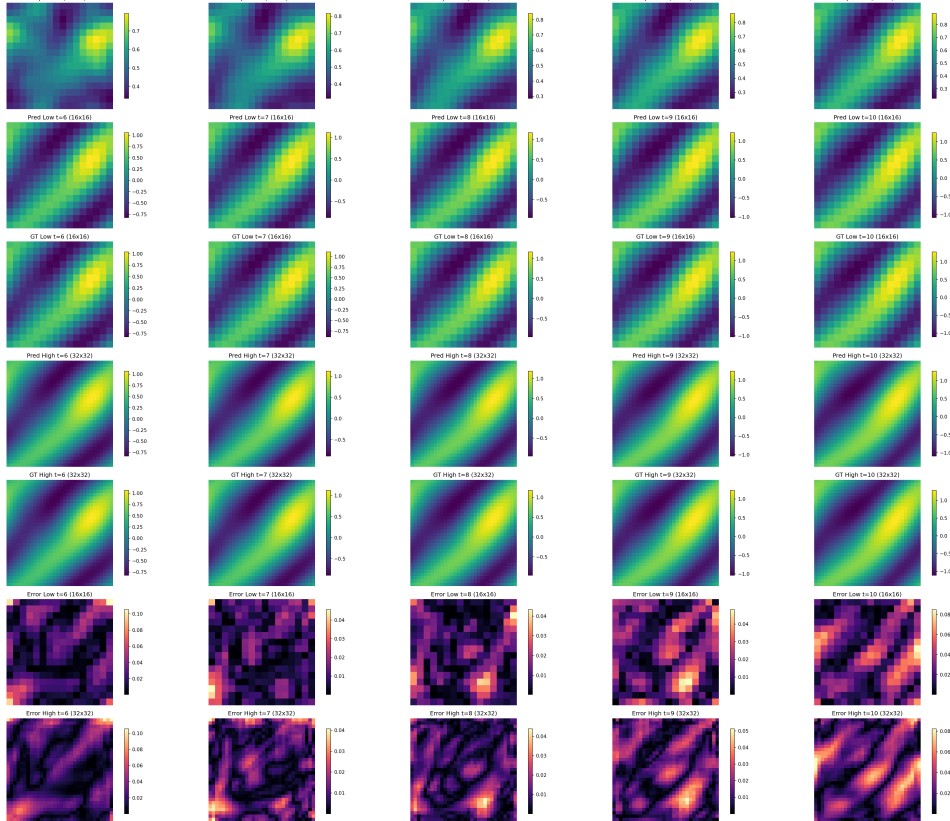

Figure 4: (Temp_DFNO) 5 steps low resolution inputs, and predictions of the model on both $16 \times 16$ and $32 \times 32$ resolutions, as well as the residuals in the last 2 rows.

### B.2  COPERNICUS OCEAN CURRENT DOWNSCALING

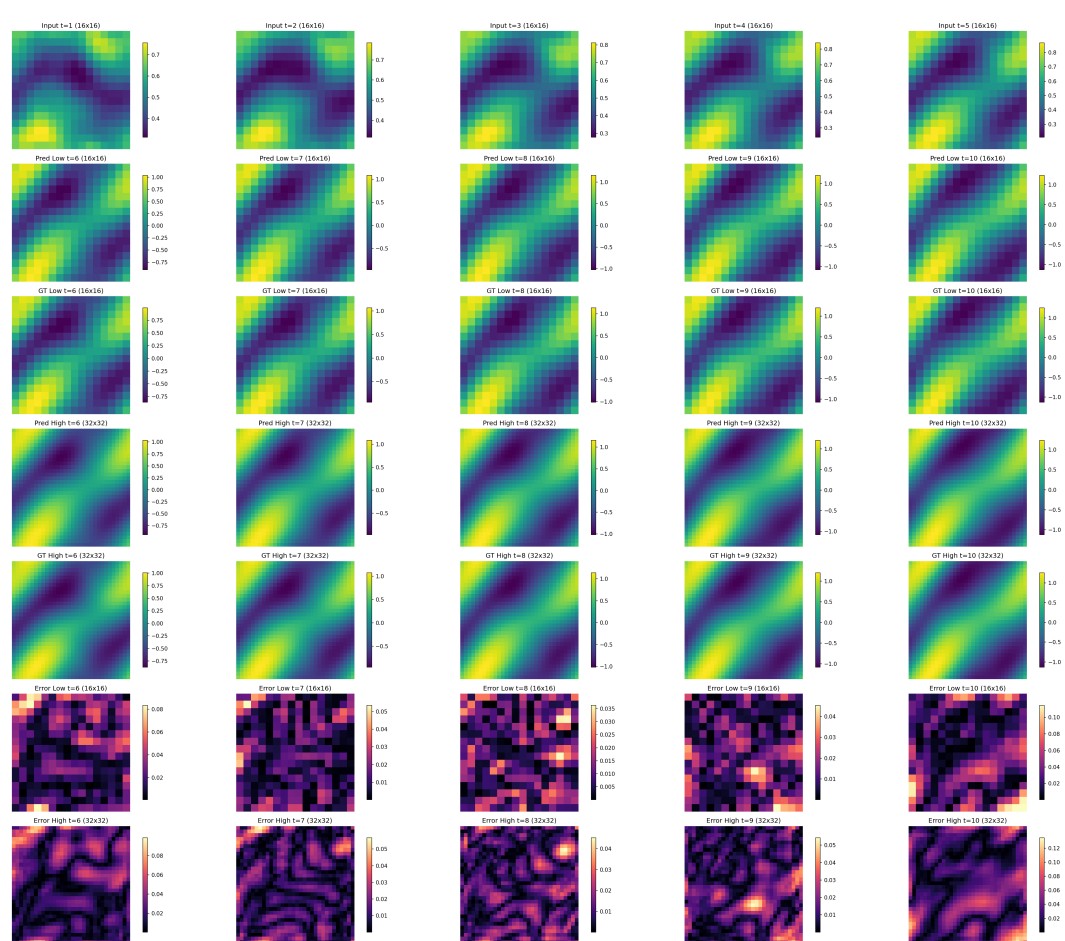

Figure 5: (Temp_specDFNO) 5 steps low resolution inputs, and predictions of the model on both $16 \times 16$ and $32 \times 32$ resolutions, as well as the residuals in the last 2 rows.

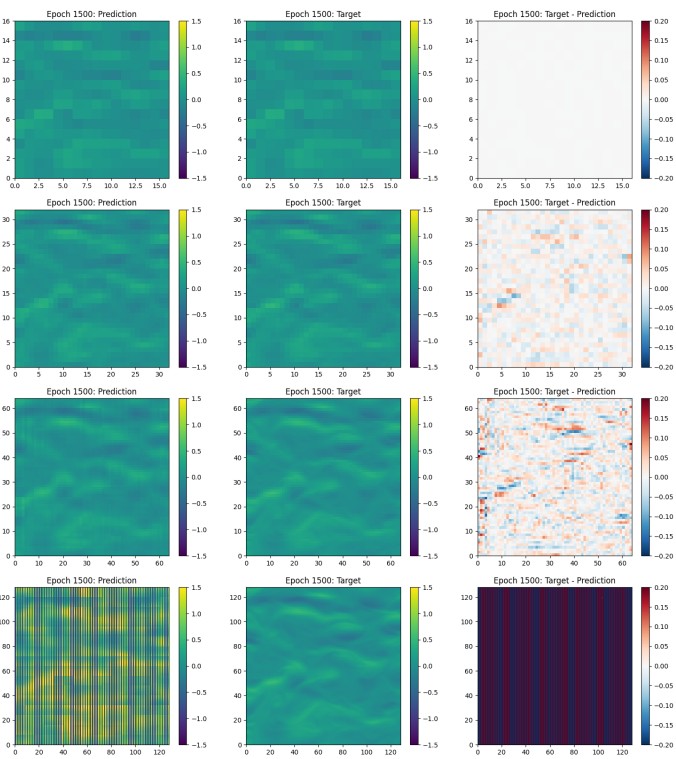

Figure 6: Ground truth vs. predictions of SpecDFNO. Rows correspond to different output resolutions: $16 \times 16$, $32 \times 32$, $64 \times 64$, and $128 \times 128$. The first column shows the model predictions, the second shows the ground truth, and the third displays the difference between them.

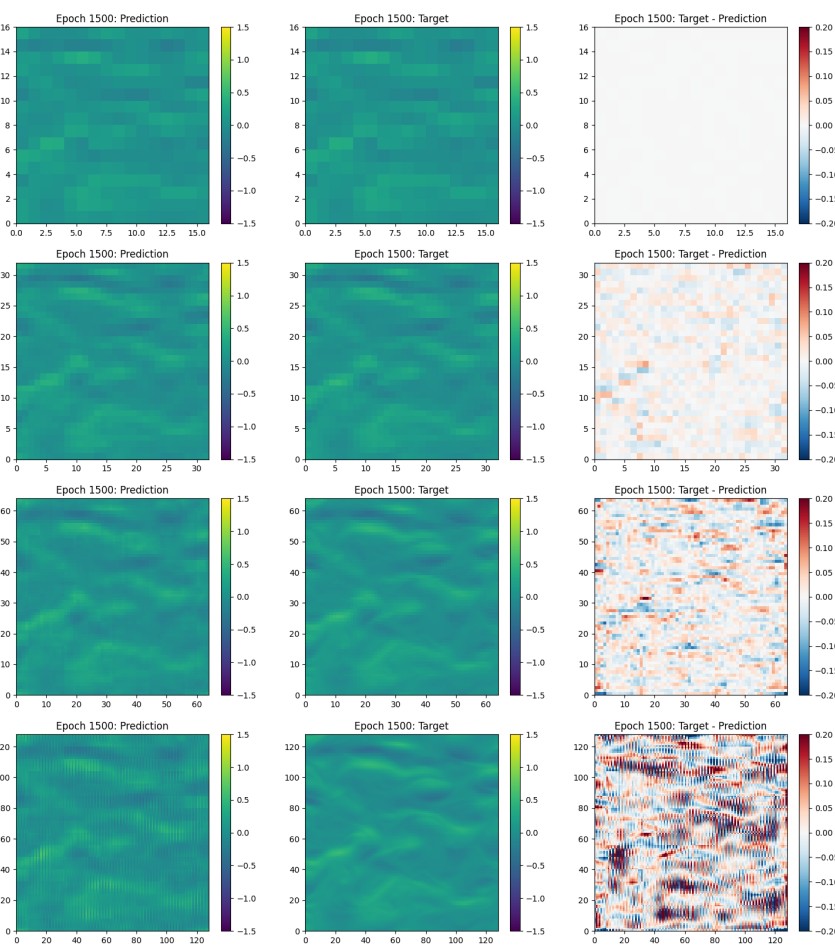

Figure 7: Ground truth vs. predictions of SpecDFNODiff. Rows correspond to different output resolutions: $16 \times 16$, $32 \times 32$, $64 \times 64$, and $128 \times 128$. The first column shows the model predictions, the second shows the ground truth, and the third displays the difference between them.

