# OpenReview forum: "Multiscale Neural PDE Surrogates for Prediction and Downscaling: Application to Ocean Currents"
_ICLR.cc/2026/Conference — ICLR 2026 Conference Desk Rejected Submission_

### Official Review · Reviewer_7bPF · 2025-10-15

**Soundness:** 1
**Presentation:** 1
**Contribution:** 1
**Rating:** 2
**Confidence:** 3

**Summary:**

This work introduces a Neural Operator-based approach for solving PDEs with arbitrary resolution solutions, and benchmarks a series of downscaling methods on ocean data.

**Strengths:**

- The writing is easy to follow and understand.

**Weaknesses:**

- The paper solely applies Neural Operator techniques on ocean data, without significant novelty on the methodology part. Proposed methods are solely combination of different techniques without strong insights and intuition.
- The related work section covers only a small portion of existing studies on super-resolution and downscaling for PDEs. Many important works are missing, especially diffusion-based approaches [1, 2]. Please also refer to [3]. Although [3] primarily focuses on climate applications, as discussed below, the proposed method should also be evaluated on climate data.
- The contributions and advantages of the proposed work are not clearly stated. While the paper introduces a series of downscaling methods, the differences between these methods and existing approaches remain unclear. Moreover, the advantages of the proposed methods are not analyzed throughout the paper.
- Although this work focuses on oceanographic applications, there appears to be no specific design tailored for oceanographic data. Such characteristics are not considered within the overall framework. Therefore, the authors should also evaluate their method on other related tasks, such as climate downscaling, to better demonstrate its practical applicability to real-world problems.
- The reported experimental results are unsatisfactory. As shown in Table 1, the performance of both `Temp_DFNO` and `Temp_SpecDFNO` is consistently worse than that of DFNO-2 under the 32×32 setting.
- The running time should also be reported as an additional metric for performance evaluation.
- For both PDE solving and downscaling tasks, only two baselines (CNN and DFNO) are considered. The authors should include more baselines, such as traditional numerical methods, earlier CNN-based approaches, recent neural operator-based super-resolution methods, and generative model-based approaches such as [1, 2]. Although some of these methods may depend on input resolution, the authors should still report comparative results to demonstrate that the proposed methods achieve competitive performance.

[1] Srivastava, Prakhar, et al. "Precipitation downscaling with spatiotemporal video diffusion." _Advances in Neural Information Processing Systems_ 37 (2024): 56374-56400.
[2] Mardani, Morteza, et al. "Residual corrective diffusion modeling for km-scale atmospheric downscaling." _Communications Earth & Environment_ 6.1 (2025): 124.
[3] Rampal, Neelesh, et al. "Enhancing regional climate downscaling through advances in machine learning." _Artificial Intelligence for the Earth Systems_ 3.2 (2024): 230066.

**Questions:**

- Are `DUNO`, `SpecDFNO`, `SpecDFNODiff`, and `MetaGradDFNO` the methods proposed by the authors? If so, the number of baselines for the downscaling task is insufficient. Additionally, including a summary table comparing these methods would help readers gain a clearer and more intuitive understanding of their differences and characteristics.
- Although the authors discuss various types of neural operators in Section 2.1, the motivation for adopting FNO is not clearly articulated. Furthermore, the paper does not explain how this work differs from _OceanNet_, as mentioned in lines 84–85.
- For the embedding function introduced in line 173, the notation $e(\cdot)$ should be expressed more rigorously.
- To comply with the standard citation style required by ICLR, please use `\citep{}` for most cases.
- The analysis described in Section 5.1.2 is missing and should be completed.
- PSNR and SSIM metrics should also be reported for DFNO-2 and DFNO-4 to ensure a fair comparison.

---

### Official Review · Reviewer_dL8b · 2025-11-01

**Soundness:** 2
**Presentation:** 1
**Contribution:** 2
**Rating:** 2
**Confidence:** 4

**Summary:**

The paper proposes a multiscale neural operator framework that unifies PDE surrogate modeling and resolution-agnostic downscaling, focusing on both synthetic Navier–Stokes data and real-world Copernicus ocean surface currents. The method generalizes Fourier Neural Operators (FNOs) by incorporating multiple architectural variants. The framework supports both temporal prediction and static downscaling tasks, allowing zero-shot prediction at arbitrary resolutions. Experiments show that the proposed models outperform U-Net CNN baselines on both synthetic and observational datasets. However, performance declines for extreme scaling. The study concludes that neural operators provide an efficient alternative to traditional numerical solvers and a practical tool for resolution-agnostic ocean modeling.

**Strengths:**

1. The paper presents a clear motivation grounded in real-world needs for high-resolution ocean current prediction, supported by a solid connection between PDE-based simulation and Earth observation data.
2. The proposed framework effectively integrates operator learning with flexible resolution control, demonstrating both theoretical generalization and practical benefits.
3. The inclusion of multiple architectural variants offers a valuable comparative perspective and highlights design trade-offs between accuracy and computational cost.

**Weaknesses:**

1. The improvements among the neural operator variants are relatively small, and it is unclear whether these differences are statistically significant.
2. While the models demonstrate strong quantitative performance, qualitative figures mostly show generic flow features; no physical validation or conservation checks are reported for real-world data.
3. The study lacks ablation analyses that would clarify which components are truly responsible for gains.
4. The experimental setup for Copernicus data is limited to static snapshots without temporal evolution.
5. Computational efficiency and scalability on large domains are not discussed, even though this is a primary motivation for neural operators.
6. Related work discussion focuses on FNO-based models but overlooks recent physics-informed and transformer-based PDE surrogates beyond Fourier domains.

**Questions:**

1. Have you examined whether the diffusion-based upsampling produces physically meaningful fine structures or just visually sharper outputs?
2. What are the main limitations when generalizing this model to other variables, such as temperature or salinity?

---

### Official Review · Reviewer_LABc · 2025-11-01

**Soundness:** 3
**Presentation:** 2
**Contribution:** 2
**Rating:** 4
**Confidence:** 3

**Summary:**

This paper tackles the downscaling (super-resolution) for PDE surrogate modeling. The paper proposes several variants of DFNO model and incorporates gradient information. The models are tested on time-dependent Navier-Stokes equation modeling and static ocean data downscaling.

**Strengths:**

- The models are tested on real-world data.
- Ablation studies against the base model show the usefulness of gradient information.

**Weaknesses:**

- The temporal modeling is only tested for the synthetic NS data, not for the ocean data.
- Some details of the models may not be clearly described, such as the base DFNO model, the reconstruction block, the constraint block in Figure 1, additional model layers in SpecDFNO, the diffusion operations in SpecDFNODiff etc.

**Questions:**

- Is downscaling the same as super-resolution.
- If the above is true, would direct interpolation (e.g., bicubic interpolation) be a baseline (no parameters)?
- Are SpecDFNO and DFNO the main models the paper wants to present? How MetaGradDFNO and MultiGradDFNO work?
-  Sec 3.4: the model predicts both low (16x16) and high (32x32) resolutions, what does this mean? Are there different targets during training?
- Font for Figure 1 is too small

---

### Note · Program_Chairs · 2026-01-17
**Submission Desk Rejected by Program Chairs**

The following references in this submission do not refer to real documents and/or have major errors in bibliographic information:

 Suhash et al. Chattopadhyay. Oceannet: Surrogate modeling of gulf stream dynamics using fno and predictor-corrector schemes. Advances in Water Resources, 2023.